# Multi-Omics Integration Analysis Pinpoint Proteins Influencing Brain Structure and Function: Toward Drug Targets and Neuroimaging Biomarkers for Neuropsychiatric Disorders

**DOI:** 10.3390/ijms25179223

**Published:** 2024-08-25

**Authors:** Yunzhuang Wang, Sunjie Zhang, Weiming Gong, Xinyu Liu, Qinyou Mo, Lujia Shen, Yansong Zhao, Shukang Wang, Zhongshang Yuan

**Affiliations:** 1Department of Biostatistics, School of Public Health, Cheeloo College of Medicine, Shandong University, 44, Wenhua West Road, Jinan 250012, China; wyz202236493@mail.sdu.edu.cn (Y.W.); zhangsunjie@mail.sdu.edu.cn (S.Z.); gongweiming@mail.sdu.edu.cn (W.G.); liuxinyu202236458@mail.sdu.edu.cn (X.L.); moqinyou@mail.sdu.edu.cn (Q.M.); shenlujia@mail.sdu.edu.cn (L.S.); 202216398@mail.sdu.edu.cn (Y.Z.); 2Institute for Medical Dataology, Shandong University, 12550, Erhuan East Road, Jinan 250003, China

**Keywords:** image-derived phenotypes, brain and plasma proteins, omics-integration analyses, pleiotropy analysis, drug targets, neuropsychiatric disorders

## Abstract

Integrating protein quantitative trait loci (pQTL) data and summary statistics from genome-wide association studies (GWAS) of brain image-derived phenotypes (IDPs) can benefit in identifying IDP-related proteins. Here, we developed a systematic omics-integration analytic framework by sequentially using proteome-wide association study (PWAS), Mendelian randomization (MR), and colocalization (COLOC) analyses to identify the potentially causal brain and plasma proteins for IDPs, followed by pleiotropy analysis, mediation analysis, and drug exploration analysis to investigate potential mediation pathways of pleiotropic proteins to neuropsychiatric disorders (NDs) as well as candidate drug targets. A total of 201 plasma proteins and 398 brain proteins were significantly associated with IDPs from PWAS analysis. Subsequent MR and COLOC analyses further identified 313 potentially causal IDP-related proteins, which were significantly enriched in neural-related phenotypes, among which 91 were further identified as pleiotropic proteins associated with both IDPs and NDs, including *EGFR*, *TMEM106B*, *GPT*, and *HLA-B*. Drug prioritization analysis showed that 6.33% of unique pleiotropic proteins had drug targets or interactions with medications for NDs. Nine potential mediation pathways were identified to illustrate the mediating roles of the IDPs in the causal effect of the pleiotropic proteins on NDs, including the indirect effect of *TMEM106B* on Alzheimer’s disease (AD) risk via radial diffusivity (RD) of the posterior limb of the internal capsule (PLIC), with the mediation proportion being 11.18%, and the indirect effect of *EGFR* on AD through RD of PLIC, RD of splenium of corpus callosum (SCC), and fractional anisotropy (FA) of SCC, with the mediation proportion being 18.99%, 22.79%, and 19.91%, respectively. These findings provide novel insights into pathogenesis, drug targets, and neuroimaging biomarkers of NDs.

## 1. Introduction

Image-derived phenotypes (IDPs) for brain structure and function are associated with many neuropsychiatric traits [1]. In particular, the association between the microstructural differences in white matter (WM) tracts and Alzheimer’s disease (AD), mild cognitive impairment, and Parkinson’s disease (PD) is well-established [2,3], so is the association between brain volume and Schizophrenia (SCZ) [4,5]. The WM tracts and brain volume are typically measured using diffusion multimodal magnetic resonance imaging (dMRI) and T1-weighted MRI (T1-MRI), respectively. Using the diffusion tensor imaging (DTI) model, several DTI parameters are exported to represent information transmission between the WM of the brain [6], and several region-of-interest (ROI) parameters in T1-MRI analysis are often used to represent brain volume [7]. Investigating the genetic architecture underlying DTI parameters and ROI volumes can enhance understanding of the genetic basis of brain structure variations and several neuropsychiatric traits.

Genome-wide association studies (GWAS) have identified variants associated with DTI parameters and ROI volumes; however, the molecular mechanism underlying these associations remains unclear. Indeed, many genetic variants could exert effects by regulating protein abundance, which are key substances related to the integrity of WM tracts and brain volume. For example, DNA methylation C-reactive protein is associated with total brain volume, gray matter volume, WM volume, regional brain atrophy, and poorer WM microstructure [8]. Translocator protein 18 kDa (*TSPO*) shows higher distribution volumes in 8 brain regions [9]. Identifying the proteins associated with DTI parameters and ROI volumes is crucial for elucidating the genetic mechanisms underlying brain structural changes and benefiting the diagnosis and treatment of neuropsychiatric disorders (NDs).

Integrating protein quantitative trait loci (pQTL) data and summary statistics from GWAS can help identify the proteins associated with DTI parameters and ROI volumes. In particular, this proteome-wide association study (PWAS) aims to assess how genetic variants affect DTI parameters and ROI volumes through proteins [10,11]. Mendelian randomization (MR) is an effective method to infer the causal effect of proteins on the outcomes. In addition, colocalization (COLOC) analysis can be utilized to identify the shared causal variants between proteins and the outcomes. Recently, MR analysis has been used to investigate the causal associations between blood biomarkers and subcortical brain structure volumes [12] and, together with COLOC analysis, between structural changes in specific brain regions and the risk of amyotrophic lateral sclerosis (ALS) [13]. However, these studies only involve a few proteins, IDPs, and NDs, failing to explore the complex relationship among them systematically. In addition, these studies mainly focus on using a single method. Inevitably, integrating the findings from multiple methods can avoid the bias from only using one method and thus improve the robustness of the results.

In this study, we developed a cutting-edge systematic analytic framework by sequentially using PWAS, MR, and COLOC analyses to identify the brain and plasma proteins causally associated with 211 IDPs (101 ROI volumes and 110 DTI parameters) as well as the implications for 15 NDs (attention deficit hyperactivity disorder (ADHD), Anorexia nervosa (AN), anxiety disorder (ANX), Autism spectrum disorder (ASD), bipolar disorder (BIP), major depressive disorder (MDD), obsessive-compulsive disorder (OCD), post-traumatic stress disorder (PTSD), SCZ, Tourette syndrome (TS), AD, ALS, Lewy body dementia (LBD), multiple sclerosis (MS), and PD). Specifically, we first conducted PWAS to screen the brain and plasma proteins associated with IDPs. Second, we performed MR analysis to investigate the causal effects of proteins identified from the PWAS analysis on IDPs. Third, we performed COLOC analysis on significant proteins from MR analysis to remove the spurious MR signals due to distinct variants with linkage disequilibrium (LD), followed by phenotype enrichment analysis to investigate whether the remaining proteins are associated with certain neural phenotypes. Fourth, we linked the IDP-associated proteins identified from the comprehensive analyses above with the 15 NDs to determine the pleiotropic proteins and performed mediation analysis to evaluate the mediating role of either DTI parameters or ROI volumes in the causal effect of these pleiotropic proteins on the NDs. Finally, the drug-gene interaction database (DGIdb) was utilized to prioritize potential drug targets. The overall study design and analysis procedure are displayed in Figure 1.

## 2. Results

### 2.1. PWAS Identified 2557 Protein-IDP Associations

We performed PWAS by integrating GWASs of IDPs (101 ROI volumes and 110 DTI parameters) with brain and plasma pQTL data, respectively. We totally identified 1635 brain protein-DTI associations involving 108 DTI parameters and 315 brain proteins with 48 (15.24%) brain proteins being associated with more than 5 DTI parameters, 711 plasma protein-DTI associations involving 95 DTI parameters and 155 plasma proteins with 19 (12.26%) plasma proteins being associated with at least 5 DTI parameters, 137 brain protein-ROI associations involving 43 ROI volumes and 83 brain proteins with 16 (19.28%) had more than 2 associations, and 74 plasma protein-ROI associations involving 39 ROI volumes and 46 plasma proteins with 15 (32.61%) plasma proteins being associated with at least 2 ROI volumes (Figure 2A,B). All results suggested that one protein can be significantly associated with multiple IDPs and vice versa. Details are provided in Appendix A.

### 2.2. MR Identified 2184 Protein-IDP Associations

For the significant associations from PWAS analysis, we, through a comprehensive MR analysis, totally identified 1452 casual associations involving 286 brain proteins and 108 DTI parameters, 540 casual associations involving 136 plasma proteins and 93 DTI parameters, 131 casual associations involving 78 brain proteins and 42 ROI volumes, and 61 casual associations involving 37 plasma proteins and 34 ROI volumes. For example, causal associations had been detected between the brain protein-coding gene *ABCG2* and radial diffusivity (RD) of the external capsule (*β* = 0.231, 95% confidence interval (*CI*): 0.108~0.354, *p*-adjust = 4.36 × 10^−4^), between the plasma protein-coding gene *ENPP6* and fractional anisotropy (FA) of the anterior corona radiata (*β* = 0.207, 95% *CI*: 0.078~0.335, *p*-adjust = 2.36 × 10^−3^), between the brain protein-coding gene *ATP13A2* and the volume of left rostral middle frontal (*β* = 0.916, 95% *CI*: 0.520~1.313, *p*-adjust = 1.97 × 10^−5^), and between the plasma protein-coding gene *TREM2* and the volume of right pars triangularis (*β* = −0.179, 95% *CI*: −0.347~−0.012, *p*-adjust = 3.69 × 10^−2^).

In addition, all *F*-statistics were larger than 10, indicating little weak instrument bias. The MR-steiger tests suggested no reverse causal association issue. Cochrane’s Q-statistic (*p-*value > 0.05) indicated no heterogeneity between IVs. The MR-Egger intercept did not identify any pleiotropic Single Nucleotide Polymorphisms (SNPs), and the leave-one-out sensitivity test also showed the robustness of the results. All details are provided in Appendix A.

### 2.3. COLOC Identified 1085 Protein-IDP Associations

COLOC analysis was further implemented for all 2184 significant protein-IDP pairs from MR analysis to identify those pairs that were unlikely to be biased by LD, with details provided in Appendix A. Briefly, we identified 758 brain protein-DTI pairs with significant evidence of COLOC involving 169 brain proteins and 105 DTI parameters, with 40 (23.67%) brain proteins colocalized with more than 5 DTI parameters; 210 plasma protein-DTI pairs with significant COLOC evidence involving 68 plasma proteins and 80 DTI parameters, with 10 (14.71%) plasma proteins colocalized with more than 5 DTI parameters; 88 brain protein-ROI pairs with significant COLOC evidence involving 56 brain proteins and 36 ROI volumes, with 17 (30.36%) brain proteins colocalized with at least 2 ROI volumes; and 29 plasma protein-ROI pairs with significant COLOC evidence involving 20 plasma proteins and 23 ROI volumes, with 5 (25.00%) plasma proteins colocalized with at least 2 ROI volumes. In summary, we identified a total of 313 COLOC-significant proteins.

Of note, we identified 30 brain proteins and 4 plasma proteins that were commonly colocalized with both DTI parameters and ROI volumes (Figure 2C), such as the colocalization between brain protein-coding gene *EIF2B3* and volume of the right precuneus, axial diffusivity (AxD) of the splenium of corpus callosum (SCC), AxD of the body of corpus callosum (BCC), and AxD of the genu of corpus callosum (GCC) (PP.H4 = 0.745, 0.920, 0.957, 0.880, respectively), between brain protein-coding gene *MSTO1* and volume of right cuneus, AxD of the uncinate fasciculus (UNC) (PP.H4 = 0.799, 0.794, respectively), and between plasma protein-coding gene *TIE1* and 6 ROI volumes and 24 DTI parameters.

### 2.4. The Phenotype Enrichment Analysis

In the phenotype enrichment analysis, we first removed 45 overlapped proteins from 313 COLOC-significant proteins, and 268 proteins remained, among which 86 proteins significantly enriched to behavior/neurological phenotype (32.09%; *p* = 3.22 × 10^−2^), and 76 proteins significantly enriched to nervous system phenotype (28.36%; *p* = 2.54 × 10^−3^). Overall, we identified 40.24% brain proteins and 41.18% plasma proteins for 110 DTI parameters, while 42.86% brain proteins and 40.00% plasma proteins for 101 ROI volumes, to be associated with at least one of these two kinds of phenotypes (Figure 2D). The detailed results are provided in Appendix A.

### 2.5. Pleiotropy Analysis with Neuropsychiatric Disorders

Among the 313 COLOC-significant proteins, 91 (79 unique) proteins were identified as the pleiotropic proteins that causally associated with both IDPs and at least one of 15 NDs, including 47 brain proteins for 87 DTI parameters and 12 NDs, 13 plasma proteins for 51 DTI parameters and 6 NDs, 26 brain proteins for 26 ROI volumes and 12 NDs, and 5 plasma proteins for 11 ROI volumes and 4 NDs (Figure 3). Significant pleiotropic proteins include the brain protein-coding gene *HLA-B* for 12 DTI parameters (dti29, dti105, dti121, etc.) and 4 NDs (BIP, MDD, MS, and SCZ), the brain protein-coding gene *TMEM106B* for 6 DTI parameters (dti25, dti39, dti319, dti337, dti354, and dti417) and 4 NDs (AD, ANX, BIP, and MDD), and the plasma protein-coding gene *GPT* for 5 DTI parameters (dti7, dti22, dti52, dti55, and dti322) and MDD. The full phenotypes of the DTI parameters and ROI volumes were provided in Appendix A, and detailed results were summarized in Appendix A.

### 2.6. Druggable Targets Exploration

We finally explored whether the 79 unique pleiotropic proteins can serve as potential therapeutic targets (Figure 3), with details provided in Appendix A. Briefly, we identified 5 protein-coding genes (*EGFR*, *GPT*, *FASN*, *ERBB3*, and *HLA-B*) that had drug targets or interactions with medications for NDs. Forty-one protein-coding genes (*PLCD3*, *GPNMB*, *KHK*, *NEK4*, *PLCG1*, *TIE1*, *PPCDC*, *RSPO3*, *RCSD1*, *TMEM106B*, etc.) may serve as potential drug targets, whose expression are associated with neurodevelopmental and brain disorders. These findings are expected to promote and facilitate the development of specific drugs for NDs.

### 2.7. Mediation Analysis

MR mediation analysis further identified 9 potential mediation pathways from the pleiotropic proteins to the corresponding NDs through IDPs (Figure 3), including 5 for AD, 2 for SCZ, 1 for BIP, and 1 for MDD. For example, we found three indirect effects of *EGFR* on AD risk via RD of the posterior limb of the internal capsule (PLIC), with the mediation proportion being 18.99%, via RD of SCC, with the mediation proportion being 22.79%, and via FA of SCC with the mediation proportion being 19.91%. All details are provided in Table 1.

## 3. Discussion

We have performed a comprehensive omics-integration analysis to identify 237 unique DTI parameter-associated proteins and 76 unique ROI volume-associated proteins, followed by a series of downstream analyses, including phenotype enrichment analysis, both pleiotropy analysis and mediation analysis with NDs, and drug prioritization analysis. We aimed to provide a deep investigation of the complex relationship among brain or plasma proteins, DTI parameters, ROI volumes, and NDs. Overall, the number of causal proteins for WM is much larger than that for brain volume, which is consistent with previous studies that WM structure could contribute more to the etiology of NDs than brain subregion volumes [14]. Indeed, WM is the tissue that transforms messages across different areas of grey matter within the central nervous system (CNS), and its integrity is critical for the brain to transmit nerve impulses [15]. Moreover, WM lesions on MRI are associated with many adverse outcomes, including cognitive impairment, functional disability, NDs, and death [2,16,17,18,19]. Thus, WM integrity may play a much broader role than the volume of brain structure in brain function.

Among the brain and plasma proteins analyzed, 67 of 237 unique DTI parameter-associated proteins and 22 of 76 unique ROI volume-associated proteins are replicated in previous GWAS of IDPs. In addition, 62 of the unverified 170 proteins for DTI parameters and 17 of the unverified 54 proteins for ROI volumes have also been potentially implicated in brain structures such as WM and cerebral regional volume. Even for the remaining 130 unique proteins, after removing the overlapped 15 proteins, 81 proteins have been found to be associated with NDs. Of note, 30 brain proteins are associated with both DTI parameter and ROI volumes, among which 11 brain protein-coding genes (*ABCG2*, *CCDC90B*, *EGFR*, *NUDT14*, *PCMT1*, *PPP2R4*, *SGTB*, *TBC1D9B*, *AMPD3*, *STX6*, and *TOM1L2*) show strong evidence from the GWAS of IDPs. In particular, *NUDT14* and *TOM1L2* are associated with whole-brain restricted directional diffusion [20], while *PCMT1*, *PPP2R4*, and *CCDC90B* are associated with brain region volumes [21,22]. Four plasma proteins are also associated with DTI parameters and ROI volumes, among which 2 plasma protein-coding genes (*NEO1*, *TIE1*) show strong evidence from the GWAS of IDPs, with *TIE1* associated with brain region volumes [23]. Among the remaining proteins, 9 brain protein-coding genes (*IGFBP2*, *EIF2B3*, *MSTO1*, *HYI*, *SLC20A2*, *UBQLN4*, *C3orf18*, *ATP13A2*, and *AP3D1*) and 1 plasma protein-coding gene (*ACADVL*) were reported to be associated with WM or brain region volume. For example, the protein-coding gene *EIF2B3* is required for myelin formation in the early stages of CNS development and is associated with leukodystrophy with vanishing WM [24]. MRI of patients with recessive mutations in *MSTO1* showed cerebellar atrophy and high intensity-signals in the ventricular WM [25,26]. Patients with *ACADVL* mutation have mild high signals in the periventricular WM, mainly in the posterior region of the head [27].

Several identified proteins not only show pleiotropic associations with the NDs but are expected to be or have emerged as clinical drug targets. For instance, we identified brain protein-coding gene *EGFR* associated with 15 IDPs (14 DTI parameters and 1 ROI volume) and AD. In an animal model of Drosophila melanogaster expressing Amyloid-β42 (a model of AD), *EGFR* inhibitor treatment ameliorates Amyloid-β42-induced memory loss [28]. Furthermore, a previous study states that *EGFR* is a potential dual molecular target for both cancer and AD [29]. CURCUMIN, a drug that interacts with *EGFR*, is used to treat mild cognitive impairment [30,31]. We identified the brain protein-coding gene *HLA-B* associated with 12 DTI parameters and 4 NDs. *HLA-B* was found to be causally associated with both DTI parameters and MS, as well as both DTI parameters and SCZ. A previous study suggests that *HLA-B*44* is a protective genetic subtype for multiple sclerosis susceptibility and improves the MRI results, such as preserved brain volume and reduced brain lesion volume [32]. In addition, common *HLA* variants are also associated with altered and asymmetrical thalamic and hippocampal volumes in SCZ [33]. The drugs CLOZAPINE and INTERFERON BETA-1A, interacting with *HLA-B*, are used clinically for the treatment of refractory SCZ and relapsing MS, respectively. *GPT* was found to be causally associated with both DTI parameters and MDD. *GPT* variants interfere with synaptic neurotransmitter release, leading to generalized CNS dysfunction, with patients presenting significant intellectual disability and MRI showing subtle incomplete and abnormal myelin formation involving subcortical WM [34]. The drug PHENELZINE targeted *GPT* is used for the treatment of atypical, non-endogenous, or neurotic depression.

In mediation analysis to investigate the potential mediating roles of the IDPs in the causal effect of the pleiotropic proteins on the NDs, we found that the effect of *EGFR* on AD can be mediated by 3 DTI parameters, including PLIC.RD, SCC.RD, and SCC.FA and the effect of *TMEM106B* on AD can be mediated by PLIC.RD. *EGFR* plays an important role in oligodendrogliogenesis and myelin production and is associated with the onset of neurodegenerative diseases such as AD, PD, and ALS [35]. *TMEM106B* is a genetic determinant of cerebrospinal fluid AD biomarker levels [36], and markers in *TMEM106B* are associated with AD through mechanisms involving neuronal injury and inflammation [37]. Significantly, AD patients have reduced FA, increased RD in SCC, and increased RD in PLIC compared to normal control [38]. The new discovery of the mediating role of IDPs not only helps to explain the mechanisms of corresponding NDs but also provides a new basis for early diagnosis and new opportunities for early treatment of corresponding NDs.

Our study has limitations. First, due to the data availability, our findings are primarily based on European individuals and cannot be directly extended to other non-European populations. Second, the small sample size of brain pQTL data leads to the insufficient number of SNPs in brain MR analysis and loss of power. Increasing the sample size could provide more robust and conclusive results. Third, we only incorporated two types of IDPs, while using multiple imaging modalities, such as task functional MRI and resting-state functional MRI, can provide a more comprehensive view of brain structure and function. Finally, we have to restrict our analysis to the cross-sectional framework due to the unavailability of longitudinal data in brain and blood proteomics studies. Although we have developed a cutting-edge systematic analytic framework sequentially using a variety of statistical genetic approaches, cross-sectional data cannot capture the progression of identified proteins.

## 4. Materials and Methods

### 4.1. Data Sources

#### 4.1.1. GWAS Summary Statistics

The GWAS summary statistics for ROI volumes were obtained from the UK Biobank (UKB) cohort [23], which included 19,629 European participants and used the standard OASIS-30 Atropos template for registration and Mindboggle-101 atlas for labeling [39]. The summary statistics for 110 DTI parameters were also obtained from the UKB cohort [6]. Briefly, the ENIGMADTI pipeline [40] was used to analyze UKB dMRI data for 33,292 European participants and generate 110 DTI parameters—namely, FA, AxD, mean diffusivity (MD), MO, and RD of 21 WM tracts as well as their mean values. We also obtained summary statistics for 10 psychiatric disorders and 5 neurodegenerative diseases with European ancestry, including ADHD (N = 225,534), AN (N = 72,517), ANX (N = 475,216), ASD (N = 46,350), BIP (N = 413,466), MDD (N = 1060,533), OCD (N = 9725), PTSD (N = 174,659), SCZ (N = 130,644), TS (N = 14,307), AD (N = 487,511), ALS (N = 138,086), LBD (N = 6618), MS (N = 115,803), and PD (N = 482,730). We notably selected the GWAS summary statistics with the largest sample size to date to ensure statistical power. All GWASs were approved by the relevant ethics committees, and written informed consent was obtained from all participants. Details are provided in Appendix A.

#### 4.1.2. Human Brain pQTL Data

The human brain proteome datasets were derived from the dorsolateral prefrontal cortex (dPFC) of postmortem brain tissues, and 376 participants were recruited by the religious orders study/memory and aging project (ROS/MAP) [41]. The isobaric tandem mass tag peptide labeling was utilized for proteome sequencing, with peptides being evaluated using liquid chromatography coupled to mass spectrometry [42]. After quality control, 8356 proteins had both proteomic and genomic data, among which 1475 proteins showing significant cis-associations with genetic variation were used for PWAS analysis. Details regarding the quality control and analysis procedure are provided in the original paper [43].

#### 4.1.3. Human Plasma pQTL Data

The plasma pQTL data was derived from a large community-based cohort study (atherosclerosis risk in communities study, ARIC), consisting of 4657 plasma proteins from 7213 individuals with European American. The relative abundance of plasma proteins or protein complexes was measured by a slow off-rate modified aptamers (SOMAmers) assay on the SomaLogic version-4 platform [44]. Each protein was first regressed on the covariates to obtain the residuals, and then the rank-inverse normalized residuals were further used to perform pQTL analysis. We finally used 1348 significant cis-heritable plasma proteins (i.e., the nonzero cis-heritability with *p-*value < 0.01) with available imputation weights for PWAS analysis. The detailed analysis procedure is provided in the original publications [45].

### 4.2. Statistical Analysis

#### 4.2.1. Proteome-Wide Association Studies

PWAS has been commonly used to explore protein-trait associations by integrating pQTL data with GWAS. Here, we conducted PWAS analysis using the FUSION pipeline with the available imputation weights of plasma proteins from the ARIC study as well as the brain proteins from the ROS/MAP datasets. Specifically, we used a built-in LD reference panel and multiple models in FUSION [46] to estimate SNP-based heritability and calculate the SNP effects on either brain or plasma proteins (weights). This way, we can select the best model and obtain the most accurate weights from the data. Further, based on the derived weights, FUSION predicts the abundance of each protein in the GWAS, followed by association analysis between the predicted protein abundance and 101 ROI volumes or 110 DTI parameters. We used the accompanying European ancestry LD reference data to minimize the influence of mismatched LD. The major histocompatibility complex (MHC) region on chromosome 6 was excluded from the analysis due to its structural complexity. We used the Benjamini–Hochberg false discovery rate (FDR) correction [47,48] for multiple testing and declared the significant PWAS associations at adjusted *p*-value < 0.05.

#### 4.2.2. Mendelian Randomization Analysis

For the PWAS-significant protein-IDP associations, we further performed MR analysis, along with a series of sensitivity analyses, to estimate the potential causal effect of each brain or plasma protein on 110 DTI parameters and 101 ROI volumes. The MR analysis conforms to the STROBE-MR Statement [49]. Given each protein has a different number of SNPs within the cis-region, we used a protein-specific Bonferroni-corrected *p*-value threshold (0.05/the number of SNPs within the cis-region) to select the instrumental SNPs and further obtained independent SNPs through LD clumping (r^2^ < 0.01 in the 1 Mb cis-region) using the 1000 Genomes European dataset as the reference panel. Then, we harmonized the exposure and outcome datasets to match the effect alleles. We also calculated *F*-statistics to evaluate the strength of IVs, with the *F*-statistics larger than 10 to indicate no weak instrument bias. The MR-Egger intercept was used to detect horizontal pleiotropy [50,51], and heterogeneity was assessed using Cochran’s Q test. We also performed the MR Steiger directionality test to evaluate whether the analysis was biased by reverse causation [52].

We used the inverse variance weighted (IVW) method as the primary MR analysis [53,54]. In addition, we applied the leave-one-out analysis to test if the removal of one IV could substantially change the causal effect estimate. Again, the *p*-values were corrected using the FDR method. All MR analyses were performed using the TwoSampleMR v.0.5.6 and MendelianRandomization v.0.6.0 package in R v.4.2.1.

#### 4.2.3. Colocalization Analysis

For the potentially causal protein-IDP associations identified from MR analysis, we further performed COLOC analysis [55] to assess whether proteins and IDPs were driven by common genetic variants, using the R package coloc 5.2.2 with default parameter settings. The COLOC analysis provided the posterior probability of five hypotheses: H0, no association with either IDPs or proteins; H1, association with IDPs but not with proteins; H2, association with proteins but not with IDPs; H3, association with both IDPs and proteins, two independent SNPs; and H4, association with both IDPs and proteins, one shared SNP. We mainly focused on the posterior probability (PP) of H4 (denoted as PP.H4) that proteins and IDPs shared one causal SNP, with a probability greater than 0.7 to indicate the significance.

#### 4.2.4. The Phenotype Enrichment Analysis

To further investigate certain biological implications of the identified proteins from COLOC analysis, we performed phenotype enrichment analysis relying on the “Mouse/Human Orthology with Phenotype Annotations” from the Mouse Genome Informatics platform (MGI, http://www.informatics.jax.org/) (accessed on 8 September 2023) [56]. To characterize phenotype specificity, we used Fisher’s exact test to examine the differences in the proportions of proteins associated with certain phenotypes in the IDP-associated proteins group against the non-associated proteins group. We mainly focused on neural-related phenotypes, including behavior/neurological phenotype and nervous system phenotype, to investigate proteins associated with at least one of these two kinds of phenotypes.

#### 4.2.5. Pleiotropy Analysis with Neuropsychiatric Disorders

For the proteins significantly associated with IDPs from COLOC analysis, we further performed MR analysis using the same procedure above to causally link these proteins with 15 NDs, including 10 psychiatric disorders (ADHD, AN, ANX, ASD, BIP, MDD, OCT, PTSD, SCZ, and TS), as well as 5 neurodegenerative diseases (AD, ALS, LBD, MS, and PD). This way, we can identify the pleiotropic proteins that are causally associated with both IDPs and NDs, which could benefit the explanations of the IDP-associated proteins from the disease perspective and provide evidence of whether the IDP-associated proteins can benefit in investigating the genetic mechanism of the NDs.

#### 4.2.6. Druggable Targets Exploration

To investigate if these pleiotropic proteins identified above can serve as targets of the existing drugs or druggable targets, we explored the interactions between these proteins (or genes) and drugs using DGIdb (version 4.0) (https://www.dgidb.org/) (accessed on 8 September 2023). DGIdb provides search and filtering of drug-gene interactions and drug genomic information. The database integrates over 30 trusted sources, containing more than 40,000 genes and 10,000 drugs, with more than 100,000 drug-gene interactions or belonging to one of 42 potential drug-gene classes, and it is widely used to prioritize potential drug targets for diseases [57].

#### 4.2.7. Mediation Analysis

For the pleiotropic proteins identified to be associated with IDPs and NDs, we further conducted the mediation analysis to investigate if the IDPs could mediate the effect of these pleiotropic proteins on the NDs. The mediation analysis was conducted within the two-step MR framework, where the indirect effect was estimated by the product of the coefficients method, and the *CI* was derived from the bootstrap approach [58].

## 5. Conclusions

Focusing on the population with European ancestry, we identified the potentially causal brain and plasma proteins for IDPs as well as the pleiotropic proteins for both IDPs and NDs, where the IDPs could mediate the effect of proteins on NDs. These findings could provide novel insights into pathogenesis, drug targets, and neuroimaging biomarkers of NDs.

## Figures and Tables

**Figure 1 ijms-25-09223-f001:**
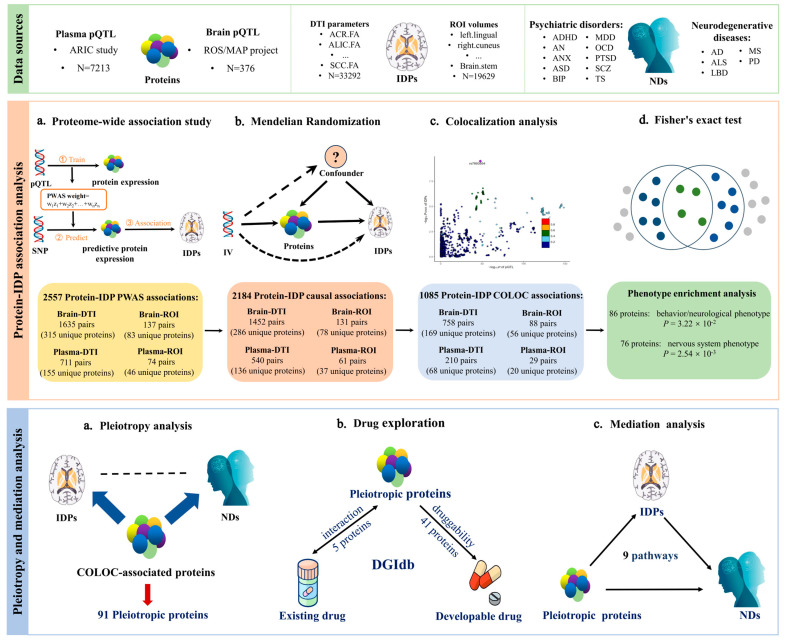
Study design. We developed a systematic omics-integration analytic framework by sequentially using ‘Protein-IDP association analysis’ (including proteome-wide association study, mendelian randomization, colocalization analysis, and Fisher’s exact test) to identify the potentially causal brain and plasma proteins for IDPs, and ‘Pleiotropy and mediation analysis’ (including pleiotropy analysis, drug exploration, and mediation analysis) to identify potential mediation pathways of pleiotropic proteins to NDs and candidate drug targets. Abbreviations: pQTL: protein quantitative trait loci; ARIC: atherosclerosis risk in communities; ROS/MAP: religious orders study/memory and aging project; DTI: diffusion tensor imaging; ACR: anterior corona radiata; ALIC: anterior limb of internal capsule; SCC: splenium of corpus callosum; ROI: region-of-interest; FA: fractional anisotropy; ADHD: attention deficit hyperactivity disorder; AN: Anorexia nervosa; ANX: anxiety disorder; ASD: Autism spectrum disorder; BIP: bipolar disorder; MDD: major depressive disorder; OCD: obsessive-compulsive disorder; PTSD: post-traumatic stress disorder; SCZ: Schizophrenia; TS: Tourette syndrome; AD: Alzheimer’s disease; ALS: Amyotrophic lateral sclerosis; LBD: Lewy body dementia; MS: multiple sclerosis; PD: Parkinson’s disease; IDPs: image-derived phenotypes; NDs: neuropsychiatric disorders; DGIdb: drug-gene interaction database.

**Figure 2 ijms-25-09223-f002:**
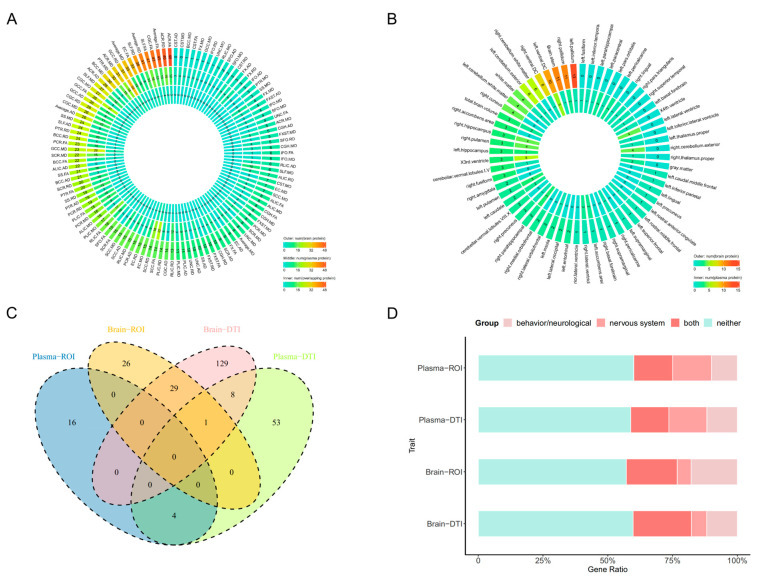
The quantitative characterization of protein-IDP association analysis. (**A**): The number of protein-DTI associations identified by PWAS analysis. The number of associations for each DTI parameter with brain and plasma proteins is presented on the outer layer and middle layer, respectively, with the number of overlapped brain and plasma proteins illustrated on the inner layer. (**B**): The number of protein-ROI associations identified by PWAS analysis. The number of associations for each ROI volume with brain and plasma proteins is presented on the outer layer and middle layer, respectively. The number of overlapped brain and plasma proteins is not shown due to fewer associations. The *p*-value < 0.05, corrected by the false discovery rate (FDR), was set as the significant threshold. (**C**): The Venn plots of significant proteins were identified after PWAS, MR, and COLOC analyses for DTI parameters and ROI volumes. (**D**): The compound bar graph of phenotype enrichment analysis for significant proteins identified after PWAS, MR, and COLOC analyses. Distinct colors represent different phenotype groups, including behavior/neurological phenotype, nervous system phenotype, both, and neither. Abbreviations: pQTL: protein quantitative trait loci; DTI: diffusion tensor imaging; ROI: region-of-interest; PWAS: Proteome-wide association study; MR: Mendelian randomization; COLOC: Colocalization.

**Figure 3 ijms-25-09223-f003:**
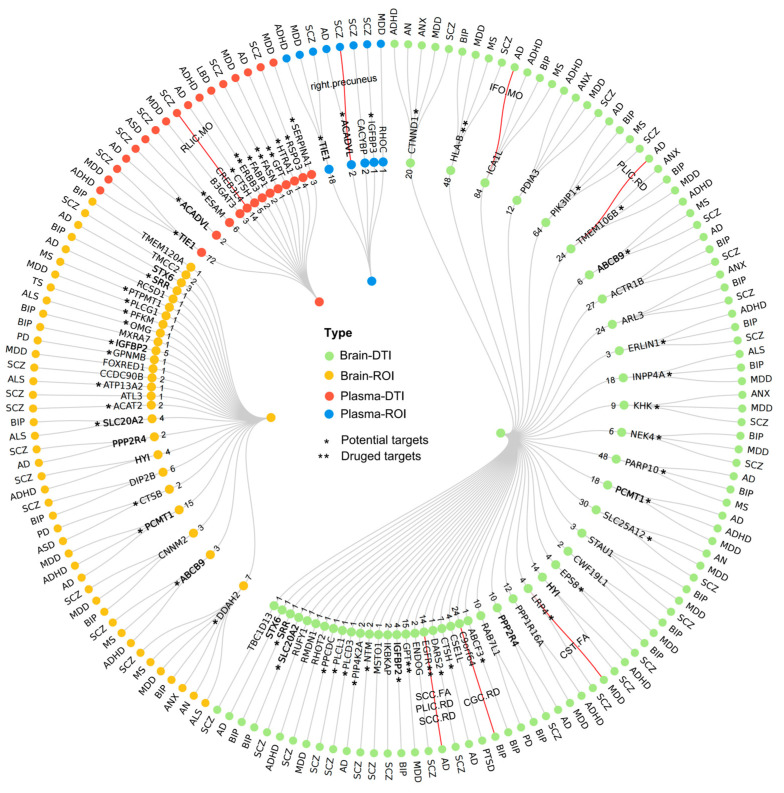
The overall landscape of the pleiotropic proteins influencing brain image-derived phenotypes and neuropsychiatric disorders. A circular dendrogram involved 79 unique pleiotropic proteins that causally associated with both brain image-derived phenotypes (IDPs) and at least one of 15 neuropsychiatric disorders (NDs), including 47 brain proteins for 87 DTI parameters and 12 NDs, 13 plasma proteins for 51 DTI parameters and 6 NDs, 26 brain proteins for 26 ROI volumes and 12 NDs, and 5 plasma proteins for 11 ROI volumes and 4 NDs. Of these, 8 brain-based genes and 2 plasma-based genes were associated with both DTI parameters and ROI volumes (marked by a bold letter). The outer layer showed NDs associated with pleiotropic proteins, the middle layer provided protein-coding genes for pleiotropic proteins, and the inner layer presented the number of IDPs associated with each pleiotropic protein. We utilized the drug-gene interaction database (DGIdb) to prioritize potential drug targets. Five unique pleiotropic proteins identified from the comprehensive analyses above have been identified as therapeutic targets for existing NDs drugs (marked by **), and 41 unique pleiotropic proteins can serve as potential therapeutic targets (marked by *). A total of 9 potential mediation pathways were identified to illustrate the mediating roles of the IDPs in the causal effect of the pleiotropic proteins on NDs in further mediation analysis. The red line between the pleiotropy protein and NDs indicates the presence of a mediating pathway. Detailed results from pleiotropy analysis (Appendix A), druggable prioritization information (Appendix A), and mediation analysis (Table 1) were also provided. Abbreviations: IDPs: image-derived phenotypes; NDs: neuropsychiatric disorders; DGIdb: drug-gene interaction database; DTI: diffusion tensor imaging; ROI: region-of-interest; RLIC: retrolenticular part of internal capsule; IFO: inferior fronto-occipital fasciculu; PLIC: posterior limb of internal capsule; SCC: splenium of corpus callosum; CGC: cingulum (cingulate gyrus); CST: corticospinal tract; MO: mode of anisotropy; RD: radial diffusivity; FA: fractional anisotropy; ADHD: attention deficit hyperactivity disorder; AN: Anorexia nervosa; ANX: anxiety disorder; ASD: Autism spectrum disorder; BIP: bipolar disorder; MDD: major depressive disorder; OCD: obsessive-compulsive disorder; PTSD: post-traumatic stress disorder; SCZ: Schizophrenia; TS: Tourette syndrome; AD: Alzheimer’s disease; ALS: Amyotrophic lateral sclerosis; LBD: Lewy body dementia; MS: multiple sclerosis; PD: Parkinson’s disease.

**Table 1 ijms-25-09223-t001:** The summary of the mediation analysis.

IDPs	Exposure	Mediator	Outcome	Total Effect	Indirect Effect (95% *CI*)	Proportion (%)
ROI	*ACADVL ^a^*	right precuneus	SCZ	−0.343	−0.187 (−0.316, −0.092)	54.54
DTI	*CREB3L4 ^a^*	RLIC.MO	SCZ	0.144	0.021 (0.009, 0.037)	14.59
	*ICA1L ^b^*	IFO.MO	AD	−0.783	−0.214 (−0.355, −0.097)	27.30
	*EGFR ^b^*	PLIC.RD	AD	0.807	0.153 (0.055, 0.270)	18.99
	*TMEM106B ^b^*	PLIC.RD	AD	0.150	0.017 (0.005, 0.032)	11.18
	*EGFR ^b^*	SCC.RD	AD	0.807	0.184 (0.087, 0.306)	22.79
	*EGFR ^b^*	SCC.FA	AD	0.807	0.161 (0.075, 0.273)	19.91
	*C9orf64 ^b^*	CGC.RD	BIP	0.424	0.096 (0.039,0.174)	22.62
	*LRP4 ^b^*	CST.FA	MDD	−0.219	−0.171 (−0.292, −0.070)	78.14

Note: *^a^* indicates plasma protein, and *^b^* indicates brain protein. Abbreviations: IDPs: image-derived phenotypes; DTI: diffusion tensor imaging; ROI: region-of-interest; RLIC: retrolenticular part of internal capsule; IFO: inferior fronto-occipital fasciculu; PLIC: posterior limb of internal capsule; SCC: splenium of corpus callosum; CGC: cingulum (cingulate gyrus); CST: corticospinal tract; MO: mode of anisotropy; RD: radial diffusivity; FA: fractional anisotropy; SCZ: Schizophrenia; AD: Alzheimer’s disease; BIP: bipolar disorder; MDD: major depressive disorder; CI: confidence interval.

## Data Availability

The datasets generated and/or analyzed during the current study are publicly available. ADHD, AN, ASD, BIP, PTSD, SCZ, TS, and OCD GWAS summary data are available from the Psychiatric Genomics Consortium database at https://pgc.unc.edu/for-researchers/download-results/ (accessed on 8 September 2023). AD, PD, ALS, and LBD GWAS summary data are available from the GWAS Catalog at https://www.ebi.ac.uk/gwas/downloads (GCST90027158, GCST009325, GCST90027164, GCST90001390) (accessed on 1 February 2024). The GWAS summary data of ANX can be downloaded at https://figshare.com/articles/dataset/Anxiety-Disorders-plink_meta_P_Sorted_gz/23659653/1 (accessed on 1 February 2024). The GWAS summary data of MDD can be downloaded at https://ipsych.dk/en/research/downloads/ (accessed on 1 February 2024). The GWAS summary data of MS is available from the MRC IEU OpenGWAS database (GWAS ID: ieu-b-18) at https://gwas.mrcieu.ac.uk/datasets/ieu-b-18/ (accessed on 1 February 2024). The GWAS summary data of 110 DTI parameters can be downloaded at https://zenodo.org/records/4549730 (accessed on 8 September 2023). The GWAS summary data of 101 ROI volumes can be downloaded at https://github.com/BIG-S2/GWAS (accessed on 8 September 2023). Plasma pQTL data can be downloaded at http://nilanjanchatterjeelab.org/pwas (accessed on 8 September 2023). Brain pQTL data can be downloaded at https://www.synapse.org/#!Synapse:syn23191787 (accessed on 8 September 2023).

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
