# Peer review of "Multi-Omics Integration Analysis Pinpoint Proteins Influencing Brain Structure and Function: Toward Drug Targets and Neuroimaging Biomarkers for Neuropsychiatric Disorders"

_ijms, 2024, doi:10.3390/ijms25179223_

Round 1

Reviewer 1 Report

Comments and Suggestions for Authors

To the AAs

Ms Summary: To elucidate the genetic mechanisms underlying brain image-derived phenotypes (IDPs) with the final aim of prevention and treating neuropsychiatric disorders (NDs), the Authors developed a systematic omics-integration analytic framework by sequentially using PWAS, MR, COLOC and phenotype enrichment analysis. Further pleiotropy analysis was needed to pinpoint common proteins associated with both IDPs and NDs, followed by mediation analysis to evaluate the role of IDPs in mediating the effect of pleiotropic proteins on NDs. A Drug-Gene Interaction Database (DGIdb) was  used to prioritize potential drug targets. A total of 10 psychiatric disorders (ADHD, AN, ANX, ASD, BIP, MDD, OCT, PTSD, SCZ, and TS), and 5 neurodegenerative diseases (AD, ALS, LBD, MS, and PD) were considered. The GWAS summary statistics for ROI volumes were obtained from the UK Biobank (UKB) cohort including 19,629 European participants. The summary statistics for 10 psychiatric disorders and 5 neurodegenerative diseases from publicly available GWAS data with European ancestry were obtained including ADHD, AN, ANX, ASD, BIP, MDD, OCD, PTSD, SCZ, TS, AD, ALS, LBD, MS, and PD. Human brain pQTL data were derived from the dorsolateral prefrontal cortex (dPFC) of postmortem brain tissues with 376 participants recruited. The plasma pQTL data derived from a large community-based cohort consisting of 4,657 plasma proteins from 7,213 individuals of European American ancestry. A total of 201 plasma proteins and 398 brain proteins associations with IDPs were detected at PWAS; further identified 313 potentially causal IDP-related proteins, significantly enriched in behavior/neurological phenotype and nervous system phenotype were identified at MR and COLOC. A little less than one third (91/313) of these potentially causal proteins were further identified to be the pleiotropic proteins linked to both IDPs and NDs. Drug prioritization analysis showed that 6.33% of unique pleiotropic proteins showed drug targets or interactions with medications for NDs. Potentially causal brain and plasma proteins for IDPs and 9 potential mediation pathways of pleiotropic proteins to NDs were identified (i.e., 5 for AD, 2 for SCZ, 1 for BIP, and 1 for MDD). The AAs’  conclusion is that their findings could provide novel insights into the pathogenesis, drug targets, and neuroimaging biomarkers of NDs. In my own opinion: the introduction provides sufficient background and include relevant references; the research design is appropriate; the methods are adequately described; the results are clearly presented; and the conclusions is sufficiently backed by the results.

Major Points

1)      Although the methodology appears solid and the results are of interest, the generalization of findings is under question given that several databases used apply to populations of European and/or European/American origin. How about populations of Asian or African ancestry? Please clarify.

2)      The Ms suffers of a rather verbose style that often negatively impacts on its readability. I would encourage the Authors to significantly reduce the words counts.

Minor Points

1)      Introduction: “While brain volume is often represented by several region-of-interest (ROI) parameters in T1- MRI analysis [8].” The sentence is incomplete, please amend.

2)      References are generally updated although, I would suggest to implement citation of the related published literature dating beyond the year 2022

3)      Although generally understandable the quality of the language needs minor to moderate improvement in style and grammar.

4)      Given the large number of acronyms, I would suggest the Authors to include an abbreviation list

Comments on the Quality of English Language

 Although generally understandable the quality of the language needs minor to moderate improvement in style and grammar.

Author Response

Comments and Suggestions for Authors: Ms Summary: To elucidate the genetic mechanisms underlying brain image-derived phenotypes (IDPs) with the final aim of prevention and treating neuropsychiatric disorders (NDs), the Authors developed a ……. In my own opinion: the introduction provides sufficient background and include relevant references; the research design is appropriate; the methods are adequately described; the results are clearly presented; and the conclusions is sufficiently backed by the results.

Responses: Thanks for your positive review, acknowledgments of our appropriate design, rigorous analysis, and presentation. Your constructive suggestions have led to substantial improvement in our manuscript. We have thoroughly addressed your comments in the revised manuscript. The detailed point-by-point responses to each of your specific comments are listed below.

Major Points

Comments 1: Although the methodology appears solid and the results are of interest, the generalization of findings is under question given that several databases used apply to populations of European and/or European/American origin. How about populations of Asian or African ancestry? Please clarify.

Responses: Thank you for your acknowledgments of our solid methods and insightful comments. Our comprehensive multi-omics integration analysis involves the data of proteome, neuroimaging, and neuropsychiatric disorders (NDs), as well as the available imputation weights of proteins. Following your comment, we have further searched the data from the GWAS Catalog, IEU OpenGWAS project, China Kadoorie Biobank, and proteomics-related literatures. Overall, for individuals with Asian ancestry, there are totally about 3414 GWAS of IDPs, and 13 GWAS of NDs, but no proteomics study. For individuals with African ancestry, there are 2 GWAS of NDs and 2 plasma protein studies, but no GWAS of IDPs. Thus, we are currently unable to perform the analysis using individuals with either Asian or African ancestry. However, as you mentioned, we do notice the importance of extending the findings to populations of other ancestries. We have highlighted in the conclusion that our results may be restricted within the European populations in the revised manuscript (line 7 on page 12). Furthermore, we have also added this limitation in the Discussion of the revised manuscript (line 25 on page 9).

Comments 2: The Ms suffers of a rather verbose style that often negatively impacts on its readability. I would encourage the Authors to significantly reduce the words counts.

Responses: Thank you for pointing this out. Following your suggestions, we have trimmed the length of the manuscript and reduced 662 words in the revision. We also carefully checked the whole manuscript again to minimize the spelling errors and to make it more concise. In addition, we have also double checked and modified the format following the journal style per the editorial requirements.

Minor Points

Comments 1: Introduction: “While brain volume is often represented by several region-of-interest (ROI) parameters in T1- MRI analysis [8].” The sentence is incomplete, please amend.

Responses: Thank you for the comment. Following your instructions, we have re-organized these sentences and have changed it to “and several region-of-interest (ROI) parameters in T1-MRI analysis are often used to represent brain volume”, to make it more clear (line 12 on page 2).

Comments 2: References are generally updated although, I would suggest to implement citation of the related published literature dating beyond the year 2022.

Responses: Thank you for the comment. Following your suggestions, we have updated the references to include relevant published literature after 2022.

Comments 3: Although generally understandable the quality of the language needs minor to moderate improvement in style and grammar.

Responses: Thank you for the suggestion. We have double checked the whole manuscript again to minimize the spelling and grammar errors as well as to modify the format following the journal style. In addition, we have also asked a native English speaker to help polish the language.

Comments 4: Given the large number of acronyms, I would suggest the Authors to include an abbreviation list.

Responses: Thank you for pointing this out. Following your suggestion, we have added an abbreviation list in the revised manuscript (line 12 on page 12), which involved a total of 48 abbreviations.

Reviewer 2 Report

Comments and Suggestions for Authors

The study integrates protein quantitative trait loci (pQTL) data and genome-wide association studies (GWAS) to identify proteins associated with brain image-derived phenotypes (IDPs) relevant to neuropsychiatric disorders (NDs). Using a comprehensive multi-omics framework, the research identifies 313 potentially causal proteins linked to brain structure and function, 91 of which are pleiotropic proteins associated with both IDPs and NDs such as Alzheimer's disease, schizophrenia, and bipolar disorder. The study also highlights nine potential mediation pathways illustrating how certain proteins influence NDs through IDPs. The findings provide insights into the genetic mechanisms of NDs and identify potential drug targets, emphasizing the role of proteins like EGFR and TMEM106B in neuroimaging biomarkers.

Please discuss or add limitation about the following 

Include a more diverse population sample to ensure the findings are applicable across different ethnicities.

The current analysis relies on a limited number of SNPs, which may lead to a loss of power. Increasing the sample size can provide more robust and conclusive results.

Using multiple imaging modalities can provide a more comprehensive view of brain structure and function, enhancing the study's depth and breadth.

Longitudinal data can help determine the stability and progression of identified biomarkers, offering insights into their potential as therapeutic targets.

Author Response

Comments and Suggestions for Authors: The study integrates protein quantitative trait loci (pQTL) data and genome-wide association studies (GWAS) ....... The findings provide insights into the genetic mechanisms of NDs and identify potential drug targets, emphasizing the role of proteins like EGFR and TMEM106B in neuroimaging biomarkers.

Responses: Thanks for your kind summary and valuable suggestions, which have led to substantial improvement of our manuscript. We have incorporated all your comments in the revised manuscript. In particular, we have added the limitations regarding the extension of the findings to other populations, discussed the issue of the limited number of SNPs as well as the potential benefits to use multiple imaging modalities and longitudinal data. The detailed point-by-point responses to each of your specific comments are listed below.

Comments 1: Include a more diverse population sample to ensure the findings are applicable across different ethnicities.

Response 1: Thank you for your insightful comments. We totally agree with you on the importance of including a more diverse population across different ethnicities, such as Asian or African ancestry. Our comprehensive multi-omics integration analysis involves the data of proteome, neuroimaging, and neuropsychiatric disorders (NDs), as well as the available imputation weights of proteins. Motivated by your suggestion, we have searched the data from the GWAS Catalog, IEU OpenGWAS project, China Kadoorie Biobank, and proteomics-related literatures, expected to obtain the data from other non-European populations. Unfortunately, except for European ancestry, there are no single ancestry that can fully cover these different levels of publically available data. For example, for individuals with Asian ancestry, there are totally about 3414 GWAS of IDPs, 13 GWAS of NDs, but no proteomics study. While for individuals with African ancestry, there are 2 GWAS of NDs, 2 plasma protein studies, but no GWAS of IDPs. Thus, we are currently unable to perform the analysis using individuals with non-European ancestries. However, as you mentioned, we do notice the importance of involving a more diverse population sample. We have highlighted in the conclusion that our results may be restricted within the European populations in the revised manuscript (line 7 on page 12). Furthermore, we have also added this limitation in the Discussion of the revised manuscript (line 25 on page 9).

Comments 2: The current analysis relies on a limited number of SNPs, which may lead to a loss of power. Increasing the sample size can provide more robust and conclusive results.

Response 2: Thank you for pointing this out. In our comprehensive analysis pipeline, the pQTL data only includes the cis-SNPs of each protein, which may potentially lead to loss of power due to the limited number of SNPs. Motivated by your comments, we further investigate the number of instrumental SNPs in cis-MR analysis using the latest summary data with relatively large sample size, including the brain pQTL data from a meta-analysis of Alzheimer’s Disease Genetics Consortium (ADGC) study (n=1277) and the plasma pQTL data from deCODE study (n=35559). In line with your comments, the median number of SNPs across 239 plasma protein-IDP pairs from deCODE study is larger than that from ARIC study used in our original manuscript (p-value = 4.2e-16 from Wilcoxon rank sum test). In addition, the median number of SNPs across 846 brain protein-IDP pairs from ADGC study is larger than that from ROS/MAP study used in our original manuscript (p-value < 2.2e-16 from Wilcoxon rank sum test). Indeed, as you mentioned, increasing sample size will provide more instrumental SNPs and thus improve the analysis power. Unfortunately, there are no publically available imputation weights in either ADGC or deCODE study, which hinders us from performing the comprehensive analysis, especially for PWAS analysis, using these two larger pQTL studies. However, we do notice the significance of your comment, and have added these statements in the Discussion of the revised manuscript (line 27 on page 9).

Comments 3: Using multiple imaging modalities can provide a more comprehensive view of brain structure and function, enhancing the study's depth and breadth.

Response 3: Thank you for the comment. We totally agree with you that using multiple imaging modalities could provide a more comprehensive view of brain structure and function and enhance the study's depth and breadth. We have added these statements in the Discussion in the revised manuscript (line 29 on page 9).

Comments 4: Longitudinal data can help determine the stability and progression of identified biomarkers, offering insights into their potential as therapeutic targets.

Response 4: Thank you for pointing this out. In this study, we are restricted with the cross-sectional design due to the unavailability of longitudinal data in brain and plasma proteomics studies. Although we have developed a cutting-edge systematic analytic framework by sequentially using a variety of statistical genetic approaches to make the results more convincing, cross-sectional data is still unable to capture the progression of identified biomarkers. We, following your suggestions, have added this limitation in the revised manuscript (line 32 on page 9).

Round 2

Reviewer 1 Report

Comments and Suggestions for Authors

To the AAs

In my personal opinion, The Authors have very carefully and satisfactorily addressed the raised criticism points. The revised version of the Ms is substantially improved in terms of both scientific communication and readability.

Reviewer 2 Report

Comments and Suggestions for Authors

No further comment